# Learning-Augmented Approximation Algorithms for Maximum Cut and Related Problems

**Vincent Cohen-Addad**
Google Research
France
coheaddad@google.com

**Tommaso d'Orsi**[*]
Bocconi University
Italy
tommaso.dorsi@unibocconi.it

**Anupam Gupta**[†]
New York University
New York NY 10012
anupam.g@nyu.edu

**Euiwoong Lee**
University of Michigan
Ann Arbor MI 48105
euiwoong@umich.edu

**Debmalya Panigrahi**
Duke University
Durham NC 27708
debmalya@cs.duke.edu

## Abstract

In recent years, there has been a surge of interest in the use of machine-learned predictions to bypass worst-case lower bounds for classical problems in combinatorial optimization. So far, the focus has mostly been on online algorithms, where information-theoretic barriers are overcome using predictions about the unknown future. In this paper, we consider the complementary question of using learned information to overcome computational barriers in the form of approximation hardness of polynomial-time algorithms for NP-hard (offline) problems. We show that noisy predictions about the optimal solution can be used to break classical hardness results for maximization problems such as the max-cut problem and more generally, maximization versions of constraint satisfaction problems (CSPs).

## 1 Introduction

The design and analysis of algorithms beyond the classical worst-case paradigm has been an active area of research (see, e.g., the collection of surveys by Roughgarden [2020]). In recent years, this has been accelerated by the success and widespread adoption of machine learning models across application domains, leading researchers to ask: *can we use machine-learned information to solve typical instances of a problem better than what we can hope for in the worst case?* This meta-question has been particularly influential in the realm of online optimization, where the goal is to design algorithms for inputs that are revealed sequentially over time. Indeed, assuming that the future unfolds in a typical rather than worst-case manner, we can compensate for the information deficit of the online algorithm with predictions learned based on past data, thereby helping it bypass information-theoretic lower bounds. This principle has been successfully applied to a broad range of online problems, such as caching [Lykouris and Vassilvitskii, 2021], rent-or-buy [Purohit et al., 2018], covering Bamas et al. [2020b], network design Azar et al. [2022], scheduling Azar et al. [2021], matching Dinitz et al. [2021], and many others (see related work for more references).

In this paper, we study the role of machine-learned predictions in *offline* NP-hard problems. For offline problems, an algorithm has no information disadvantage compared to an optimal solution: the disadvantage is computational. The NP-hardness of problems makes exact algorithms that run in polynomial time unlikely. This has led to the field of approximation algorithms, where the goal is to obtain polynomial-time algorithms that output solutions that are approximately optimal. In

---

[*]Part of this work was done when the author was at ETH Zurich.
[†]Part of this work was done when the author was at Carnegie Mellon University.

38th Conference on Neural Information Processing Systems (NeurIPS 2024).

particular, an approximation algorithm for an optimization problem has an approximation factor of $\alpha$ if the solution it produces on every instance is within a factor of $\alpha$ of that of an optimal solution. The best approximation factor for an NP-hard offline problem is also subject to computational barriers. For instance, for the classical MAXCUT problem, it is known that the approximation factor of $\alpha_{\text{GW}} \approx 0.878$ obtained in the celebrated work of Goemans and Williamson [1995] is the best possible under the Unique Games Conjecture (UGC). This raises a natural question: *can we use machine-learned predictions to overcome computational barriers to approximation algorithms for NP-hard problems?*

## 1.1 Our Contributions

Suppose we are given a noisy prediction that is mildly correlated with an optimal solution for a given problem instance. Can we use such a prediction to recover a better approximation to the optimal solution than is possible without any prediction? For (strongly) NP-hard problems, we typically know barriers for the best approximation factor $\alpha$ that can be achieved by polynomial-time algorithms. Using a machine-learned prediction of an optimal solution, we seek to go beyond this barrier: to obtain a polynomial-time algorithm with an approximation factor strictly better than $\alpha$. The quantum of improvement naturally depends on the quality of the prediction: if the prediction is $\varepsilon$-correlated with the target solution, can we get an $\alpha + f(\varepsilon)$ approximation?

To make these questions concrete, we first consider the MAXCUT problem in the learning-augmented setting. Given an undirected, weighted graph, the MAXCUT problem asks for a bi-partition of the vertices such that the total weight of edges in the cut is maximized. Assuming the widely-accepted unique games conjecture, the Goemans-Williamson approximation bound of $\alpha_{\text{GW}} \approx 0.878$ is the best possible for a polynomial-time algorithm. But, suppose we are given a prediction for the optimal max-cut that is independently correct for every vertex with probability $1/2 + \varepsilon$, for any $\varepsilon > 0$. (Note that random guessing achieves correctness of $1/2$; so, we assume that the prediction is only $\varepsilon$-better than random guesses.) Can this prediction be used to obtain an approximation factor better than $\alpha_{\text{GW}}$ in polynomial time? This question was posed by Svensson in his SODA 2023 plenary lecture.

Our first result is to unconditionally improve on worst-case performance for the MAXCUT problem using an $\varepsilon$-correlated prediction. This answers the question posed by Svensson in the affirmative. Specifically, we give an algorithm that obtains an $(\alpha_{\text{GW}} + \tilde{\Omega}(\varepsilon^4))$-approximate MAXCUT solution. This also quantifies the dependence between the improvement in the approximation factor and the correlation of the prediction with the actual optimal solution. Interestingly, we also relax the independence requirement to just pairwise independence of the predictions on the vertices. This is significant because in practice, the predictions for the different vertices are likely to be obtained from a machine learning model or a human expert, either of which sources are unlikely to output completely independent predictions for different vertices.

We further complement this result by considering another natural prediction model where instead of a noisy prediction for every vertex, we get a correct prediction but only for an $\varepsilon$-fraction of randomly chosen vertices. (To distinguish between these models, we call the former *noisy predictions* and the latter *partial predictions*.) In this case, we obtain an $(\alpha_{\text{RT}} + \Omega(\varepsilon))$-approximate solution to MAXCUT, where $\alpha_{\text{RT}} \simeq 0.858$ is the approximation factor obtained by Raghavendra and Tan for the MAXBISECTION problem [Raghavendra and Tan, 2012]. Note that $\alpha_{\text{RT}}$ is slightly smaller than $\alpha_{\text{GW}}$, but we get a better advantage of $\Omega(\varepsilon)$ instead of $\Omega(\varepsilon^4)$.

Next, we show that our algorithmic framework is applicable beyond the MAXCUT problem. *Constraint Satisfaction Problems (CSPs)* are a broad class of optimization problems that includes fundamental optimization tasks such as MAX-$k$-SAT, MAX-$k$-LIN, MAX-$k$-AND, etc. In particular, 2-CSPs are a subclass of CSPs where each constraint contains only two variables. This includes problems such as MAXCUT, MAXDICUT, and MAX-2-SAT. A classical result of Arora et al. [1999] showed that it is possible to obtain an approximation factor arbitrarily close to 1 for "dense" instances of all 2-CSPs including MAXCUT. We show that $\varepsilon$-correlated predictions of the optimal solution is a useful tool for dense instances of all 2-CSPs. In particular, we use the prediction to lower the "density threshold" for obtaining an arbitrarily small approximation factor for dense instances of 2-CSPs. In other words, the assumption on the density of the instance (which is a function of the prediction bias and the approximation error) for our result in the learning-augmented setting is weaker than that of Arora et al. [1999], i.e., our result applies to a broader set of instances.

## 1.2 Related Work

In recent years, the abundance of data and the tremendous success of machine learning has led to a variety of attempts at going beyond traditional worst-case analysis for combinatorial optimization by taking advantage of learned information. In clustering, Ashtiani et al. [2016] introduced a setting where an algorithm can query an external oracle (e.g., a machine learning model) to learn if a pair of points are in the same cluster in the optimal clustering (same-cluster queries). The goal then is to recover an optimal solution (or a sufficiently good approximation) while minimizing the number of oracle queries. Tight bounds have been obtained for various clustering objectives in this setting, from $k$-means [Ailon et al., 2018] to correlation clustering [Mazumdar and Saha, 2017]. Moreover, robust settings that incorporate noise in the oracle answers have also been studied Larsen et al. [2020], Del Pia et al. [2022]. Another line of work Ergun et al. [2022], Gamlath et al. [2022] considers $k$-means and related clustering problems where the algorithm is provided noisy node labels. For instance, Gamlath et al. [2022] showed that even when the labels provided by the oracle are correct with a tiny probability (say 1%), it is possible to obtain a $(1 + o(1))$-approximation to the $k$-means objective as long as the clusters are not too small.

A different line of work has aimed to incorporate machine-learned predictions in the design of online algorithms (see, e.g., the surveys Mitzenmacher and Vassilvitskii [2020, 2022]). This model was introduced by Lykouris and Vassilvitskii for the caching problem Lykouris and Vassilvitskii [2021] and has since been studied in many problem domains such as rent or buy Purohit et al. [2018], Gollapudi and Panigrahi [2019], Anand et al. [2020], covering Bamas et al. [2020b], Anand et al. [2022], Gupta et al. [2022], scheduling Purohit et al. [2018], Wei and Zhang [2020], Bamas et al. [2020a], Lattanzi et al. [2020], Mitzenmacher [2020], Azar et al. [2021], Cohen and Panigrahi [2023], Lindermayr et al. [2023], Lassota et al. [2023], caching Lykouris and Vassilvitskii [2021], Wei [2020], Jiang et al. [2022b], Bansal et al. [2022], matching Dinitz et al. [2021], Lavastida et al. [2021], secretary problems Antoniadis et al. [2020b], Dütting et al. [2021], graph problems Antoniadis et al. [2020a], Jiang et al. [2022a], Almanza et al. [2021], Antoniadis et al. [2023], Bernardini et al. [2022], Anand et al. [2022], Fotakis et al. [2021], Azar et al. [2022], and many others. (The reader is referred to ALP [2023] for a compendium of papers in this area.) The main goal in this line of work is to overcome information-theoretic lower bounds for online problems using machine-learned predictions about the unknown future.

Two recent works (concurrent and independent of ours) have considered the computational complexity of MAXCUT and CSPs in the context of noisy predictions. The first of these is Bampis et al. [2024], who tackle the problem of speeding up approximation schemes for dense CSP instances using noisy predictions. Namely, they provide an algorithm that achieves a $(1 - \varepsilon)$ approximation whose running time depends on the density of the instance and the error in the prediction. The algorithm runs in polynomial time if the number of edges in the instance is $\Omega(n^2/\log n)$, assuming the prediction label of each vertex is correct with constant probability. The second work is by Ghoshal et al. [2024], who consider both the partial and the noisy predictions models under full independence of the predictions. They provide a $(1 - O((\varepsilon\sqrt{\Delta})^{-1}))$-approximation for both models, where $\Delta$ is the average degree of the graph. Alternatively, they show that if the edges are weighted, the value of the solution computed is at least $\mathrm{opt} - \sqrt{n\sum_{ij} w_{ij}^2}\,\varepsilon^{-1}$.

## 2 Preliminaries

**The MAXCUT Problem.** We start by describing the MAXCUT problem. In this problem, we are given a weighted graph $G = (V, E)$ represented by a (symmetric) $n \times n$ adjacency matrix $A$, where $A_{ij} = w_{ij}$, the weight of edge $\{i, j\}$ if it exists, and 0 otherwise. (We assume the graph has no self-loops, and hence $A$ has zeroes on the diagonal.) We let $W_i = \sum_j w_{ij}$ denote the weighted degree of vertex $i$. We use $D := \mathrm{diag}(W_1, \ldots, W_n)$ to denote the diagonal matrix with these weighted degrees, and $L = D - A$ to denote the (unnormalized) Laplacian matrix of the graph. Note that $x \in \{-1, 1\}^n$ denotes a cut in the graph, and the quadratic form $\langle x, Lx \rangle = \sum_{\{i,j\} \in E} w_{ij}(x_i - x_j)^2$ counts (four times) the weight of edges crossing the cut between the vertices labeled 1, and those labeled $-1$.

The MAXCUT problem asks for the cut with maximum edge weight. Hence, MAXCUT can be rephrased as follows:
$$\text{MAXCUT}(G) := \max_{x \in \{-1,1\}^n} 1/4 \cdot \langle x, Lx \rangle.$$

We defer the formal definition of CSPs to Section 5.

**The Noisy/Partial Predictions Framework.** We describe the prediction models for the MAXCUT problem. We extend the noisy predictions model to general Max-2-CSPs in Section 5.

1. In the *noisy predictions model* for MAXCUT, we assume there is some fixed and unknown optimal solution $x^* \in \{-1, 1\}^n$. The algorithm has access to a *prediction vector* $Y \in \{-1, 1\}^n$, such that for each vertex $i$, $Y_i$ is the correct label $x_i^*$ with some (unknown) probability $1/2 + \varepsilon$, and is the other (incorrect) label with probability $1/2 - \varepsilon$. Here we only assume pairwise independence; for any two vertices $i$ and $j$, $\Pr[i, j$ both give their correct labels$] = (1/2 + \varepsilon)^2$.

2. In the *partial predictions model* for MAXCUT, the algorithm has access to a *prediction vector* $Y \in \{-1, 0, 1\}^n$ such that for each vertex $i$, $Y_i$ is the correct label $x_i^*$ with some (unknown) probability $\varepsilon$ and is 0 otherwise. Again, we only assume pairwise independence; for any two vertices $i$ and $j$, $\Pr[i, j$ both give their correct labels$] = \varepsilon^2$.

## 3 MAXCUT in the Noisy Prediction Model

Recall that in the noisy prediction model, the predicted label of each vertex is its correct label in a fixed max-cut with probability $1/2 + \varepsilon$. Moreover, the labels are pairwise independent. One can show that in this model, if one were to simply output the prediction itself, then its value would be at least $O(m/2 + \varepsilon^2(\text{opt} - m/2))$ for an unweighted graphs with $m$ edges. But, this is generally worse than the $\alpha_{GW} \cdot \text{opt} \approx 0.878 \cdot \text{opt}$ bound obtained by the Goemans-Williamson MAXCUT algorithm. Our main result is to give an algorithm that uses a noisy prediction to outperform $\alpha_{GW}$ in the approximation bound by an additive $\text{poly}(\varepsilon)$ factor:

**Theorem 3.1** (Noisy Predictions). *Given noisy predictions with a bias of $\varepsilon$, there is a polynomial-time randomized algorithm that obtains an approximation factor of $\alpha_{GW} + \tilde{\Omega}(\varepsilon^4)$ in expectation for the* MAXCUT *problem.*

The rest of this section is devoted to proving the above theorem. A basic distinction that we will use throughout this section is that of $\Delta$-wide and $\Delta$-narrow graphs; these should be thought of as weighted analogs of high-degree and low-degree graphs. We first define these and related concepts below, then we present an algorithm for the MAXCUT problem on $\Delta$-wide graphs in §3.1, followed by the result for $\Delta$-narrow graphs in §3.2. We finally wrap up with the proof of Theorem 3.1.

We partition the edges incident to vertex $i$ into two sets: the $\Delta$-*prefix* for $i$ comprises the $\Delta$ heaviest edges incident to $i$ (breaking ties arbitrarily), while the remaining edges make up the $\Delta$-*suffix* for $i$. We fix a parameter $\eta \in (0, 1/2)$. We will eventually set $\Delta = \Theta(1/\varepsilon^2)$ and $\eta$ to be an absolute constant. Recall that $W_i = \sum_{j \in [n]} A_{ij}$ is the weighted degree of $i$.

**Definition 3.2** ($\Delta$-Narrow/Wide Vertex). *A vertex $i$ is $\Delta$-wide if the total weight of edges in its $\Delta$-prefix is at most $\eta W_i$, and so the weight of edges in its $\Delta$-suffix is at least $(1 - \eta)W_i$. Otherwise, the vertex $i$ is $\Delta$-narrow.*

Intuitively, a $\Delta$-wide vertex is one where most of its weighted degree is preserved even if we ignore the $\Delta$ heaviest edges incident to the vertex.

We partition the vertices $V = [n]$ into the $\Delta$-*wide* and $\Delta$-*narrow* sets; these are respectively denoted $V_{>\Delta}$ and $V_{<\Delta}$. We define $W_{>\Delta} := \sum_{i \in V_{>\Delta}} W_i$ and $W_{<\Delta} := \sum_{i \in V_{<\Delta}} W_i$, and hence the sum of weighted degrees of all vertices is $W := \sum_{i=1}^n W_i = W_{>\Delta} + W_{<\Delta}$.

**Definition 3.3** ($\Delta$-Narrow/Wide Graph). *A graph is $\Delta$-wide if the sum of weighted degrees of $\Delta$-wide vertices accounts for at least $1 - \eta$ fraction of that of all vertices; i.e., if $W_{>\Delta} \geq (1-\eta)W$. Otherwise, it is $\Delta$-narrow.*

## 3.1 Solving MAXCUT for $\Delta$-wide graphs

For $\Delta$-wide graphs, we show:

**Theorem 3.4.** *Fix $\varepsilon' \in (0,1)$. Given noisy predictions with bias $\varepsilon$, there is a polynomial-time randomized algorithm that, given any $\Delta$-wide graph, outputs a cut of value at least the maximum cut minus $(5\eta + 2\varepsilon')W$, where $\Delta := O(1/(\varepsilon \cdot \varepsilon')^2)$, with probability $0.98$.*

Since the graph is $\Delta$-wide, most vertices have their weight spread over a large number of their neighbors. In this case, the prediction vector allows us to obtain a good estimate $\hat{r}$ of the optimal neighborhood imbalance $r^*$ (the difference between the number of neighbors a vertex has on its side versus the other side of the optimal cut). We can then write an LP to assign fractional labels to vertices that maximize the cut value while remaining faithful to these estimates $\hat{r}$; finally rounding the LP gives the solution.

### 3.1.1 The $\Delta$-wide Algorithm

Define an $n \times n$ matrix $\tilde{A}$ from the adjacency matrix $A$ as follows: for each row corresponding to the edges incident to a vertex $i$, we set the entry $\tilde{A}_{ij} = 0$ if the edge $(i,j)$ is in the $\Delta$-prefix of vertex $i$; otherwise, $\tilde{A}_{ij} = A_{ij}$. Now, define an $n$-dimensional vector $\hat{r}$ as follows:

$$
\hat{r}_i = \begin{cases} \frac{1}{2\varepsilon}(\tilde{A}Y)_i & \text{if } i \text{ is } \Delta\text{-wide} \\ 0 & \text{if } i \text{ is } \Delta\text{-narrow} \end{cases}
$$

where $Y$ is the prediction vector. Solve the linear program:

$$
\min_{x \in [-1,1]^n} \langle \hat{r}, x \rangle \quad s.t. \quad \|\hat{r} - Ax\|_1 \le (\varepsilon' + 2\eta)W. \tag{1}
$$

Let $\hat{x} \in [-1,1]^n$ be the optimal LP solution.

Finally, do the following $O(1/\eta)$ times independently, and output the best cut $X^*$ among them: randomly round the fractional solution $\hat{x}$ independently for each vertex to get a cut $X \in \{-1,1\}^n$; namely, $\Pr[X_i = 1] = (1+\hat{x}_i)/2$ and $\Pr[X_i = -1] = (1-\hat{x}_i)/2$.

### 3.1.2 The Analysis

For a labeling $x \in \{-1,1\}^n$, the *neighborhood imbalance* for vertex $i$ is defined as $\sum_j A_{ij}x_j = (Ax)_i$. This denotes the (signed) difference between the total weight of edges incident to $i$ that appear and do not appear in the cut defined by the labeling $x$. The maximality of the optimal cut $x^* \in \{-1,1\}^n$ ensures that $x_i^* \cdot \text{sign}((Ax^*)_i) \le 0$ for all $i$; else, switching $x_i$ from $1$ to $-1$ or vice-versa increases the objective. Define $r^* := Ax^*$ as the vector of imbalances for the optimal cut.

**Lemma 3.5.** *The vector $\hat{r}$ satisfies*

$$
\mathbb{E}\left[\|\hat{r} - r^*\|_1\right] := \mathbb{E}\left[\sum_{i=1}^n |\hat{r}_i - r_i^*|\right] \le O\left(\frac{W}{\varepsilon\sqrt{\Delta}}\right) + 2\eta W.
$$

*Proof.* Observe that

$$
\mathbb{E}[Y_i] = x_i^* \cdot \Pr[Y_i = x_i^*] - x_i^* \cdot \Pr[Y_i = -x_i^*] = x_i^*(1/2 + \varepsilon) - x_i^*(1/2 - \varepsilon) = 2\varepsilon x_i^*.
$$

Define $Z := \frac{1}{2\varepsilon}Y$. Then, $\mathbb{E}[Z] = x^*$, and so $\mathbb{E}[AZ] = r^*$.

First, we consider a $\Delta$-narrow vertex $i$. Since $\hat{r}_i = 0$, we have $|\hat{r}_i - r_i^*| = |r_i^*| \le W_i$. So summing over all $\Delta$-narrow vertices gives

$$
\sum_{i \in V_{<\Delta}} |\hat{r}_i - r_i^*| \le \sum_{i \in V_{<\Delta}} W_i \le \eta W, \tag{2}
$$

since the graph is $\Delta$-wide.

Now, we consider a $\Delta$-wide vertex $i$. We have

$$
|\hat{r}_i - r_i^*| = |(\tilde{A}Z)_i - r_i^*| \le |\mathbb{E}[(\tilde{A}Z)_i] - r_i^*| + |(\tilde{A}Z)_i - \mathbb{E}[(\tilde{A}Z)_i]|. \tag{3}
$$

To bound the first term in the RHS of (3), recall that $r_i^* = \mathbb{E}[(AZ)_i]$. Thus,

$$|\mathbb{E}[(\tilde{A}Z)_i] - r_i^*| = |\mathbb{E}[(\tilde{A}Z)_i] - \mathbb{E}[(AZ)_i]| = \langle (\tilde{A} - A)_i, \mathbb{E}[Z] \rangle.$$

Since $\mathbb{E}[Z] = x^* \in \{-1, 1\}^n$, we get

$$|\mathbb{E}[(\tilde{A}Z)_i] - r_i^*| = (\tilde{A} - A)_i \cdot x^* \leq \|(\tilde{A} - A)_i\|_1 \|x^*\|_\infty \leq \eta W_i,$$

where in the last step, we used the fact that $i$ is a $\Delta$-wide vertex.

Now, we bound the second term in the RHS of (3). Using Chebyshev's inequality on the sum $(\tilde{A}Z)_i = \sum_j \tilde{A}_{ij} Z_j$, we get

$$\Pr[|(\tilde{A}Z)_i - \mathbb{E}[(\tilde{A}Z)_i]| \geq \lambda_i] \leq \frac{\mathrm{var}((\tilde{A}Z)_i)}{\lambda_i^2}.$$

Since the variables $Z_j$ are pairwise independent, the variance $\mathrm{var}((\tilde{A}Z)_i) = \sum_j \tilde{A}_{ij}^2 \mathrm{var}(Z_j)$. The variance of each $Z_j$ is $O(1/\varepsilon^2)$. For $\sum_j \tilde{A}_{ij}^2$, we know

$$\sum_{j \in [n]} \tilde{A}_{ij}^2 = \|\tilde{A}_i\|_2^2 \leq \|\tilde{A}_i\|_1 \cdot \|\tilde{A}_i\|_\infty.$$

Note that the weight of any edge in the $\Delta$-suffix of $i$ is at most $W_i/\Delta$. Therefore, by our definition of $\tilde{A}$, we have $\|\tilde{A}_i\|_\infty \leq W_i/\Delta$. Since $\tilde{A}_{ij} \leq A_{ij}$ for all $j \in [n]$, we also have $\|\tilde{A}_i\|_1 \leq \|A_i\|_1 = W_i$. Applying these bounds, we get: $\sum_{j \in [n]} \tilde{A}_{ij}^2 \leq W_i^2/\Delta$. Therefore,

$$\mathbb{E}[|(\tilde{A}Z)_i - \mathbb{E}[(\tilde{A}Z)_i]|] \leq \sqrt{\mathrm{var}((\tilde{A}Z)_i)} \leq O(W_i/(\varepsilon\sqrt{\Delta})).$$

Summing over all $\Delta$-wide vertices, we get

$$\mathbb{E}\left[ \sum_{i \in V_{>\Delta}} |\hat{r}_i - r_i^*| \right] \leq O\left( \frac{W_{>\Delta}}{\varepsilon\sqrt{\Delta}} \right) + \eta W_{>\Delta} \leq O\left( \frac{W}{\varepsilon\sqrt{\Delta}} \right) + \eta W.$$

Combining with (2) for $\Delta$-narrow vertices, we get

$$\mathbb{E}\left[ \|\hat{r}_i - r_i^*\|_1 \right] \leq O\left( \frac{W}{\varepsilon\sqrt{\Delta}} \right) + 2\eta W. \qquad \square$$

Now using Markov's inequality with Lemma 3.5, we get that setting $\Delta = \Omega(1/(\varepsilon\varepsilon')^2)$ for any fixed constant $\varepsilon' > 0$ ensures that we get a vector of empirical imbalances $\hat{r}$ satisfying

$$\|\hat{r} - r^*\|_1 \leq (\varepsilon' + 2\eta)W. \tag{4}$$

with probability at least $0.99$. (Since the $2\eta W$ losses are deterministically bounded, we can use Markov's inequality only on the random variable $\sum_{i \in V_{>\Delta}} |(\tilde{A}Z)_i - \mathbb{E}[(\tilde{A}Z)_i]|$.) Hence, when the event in (4) happens, the vector $x^*$ is a feasible solution to LP (1).

Next, we need to analyze the quality of the cut produced by randomly rounding the solution of LP (1). Recall that for the (unnormalized) Laplacian $L$ and some $x \in \{-1, 1\}^n$, the cut value is

$$f(x) := 1/4 \cdot \langle x, Lx \rangle = 1/4 \cdot (\langle x, Dx \rangle - \langle x, Ax \rangle) = 1/4 \cdot (W - \langle x, Ax \rangle). \tag{5}$$

**Lemma 3.6.** *For any $\Delta$-wide graph, the algorithm from §3.1.1 outputs $X^* \in \{-1, 1\}^n$ that satisfies*

$$f(X^*) \geq f(x^*) - (2\varepsilon' + 5\eta)W$$

*with probability at least* $0.98$.

*Proof.* Recall that the cut $X^*$ is the best among $T := O(1/\eta)$ independent roundings of cut $\hat{x}$. Consider one of the roundings $X$, and write:

$$\langle X, AX \rangle = \langle \hat{x}, \hat{r} \rangle + (\langle \hat{x}, A\hat{x} \rangle - \langle \hat{x}, \hat{r} \rangle) + (\langle X, AX \rangle - \langle \hat{x}, A\hat{x} \rangle). \tag{6}$$

Let us first bound the expectation of each of the terms in the RHS of (6) separately.

To bound the first term $\langle \hat{x}, \hat{r} \rangle$, note that given (4) (which happens with probability 0.99), the solution $x^*$ is feasible for the LP in (1). This means the optimal solution $\hat{x}$ has objective function value

$$\langle \hat{r}, \hat{x} \rangle \leq \langle \hat{r}, x^* \rangle = \langle r^*, x^* \rangle + \langle \hat{r} - r^*, x^* \rangle \leq \langle x^*, Ax^* \rangle + \|\hat{r} - r^*\|_1 \|x^*\|_\infty$$
$$\leq \langle x^*, Ax^* \rangle + (\varepsilon' + 2\eta)W. \tag{7}$$

Next, we bound the second term $(\langle \hat{x}, A\hat{x} \rangle - \langle \hat{x}, \hat{r} \rangle)$ by

$$\|\hat{x}\|_\infty \|A\hat{x} - \hat{r}\|_1 \leq (\varepsilon' + 2\eta)W, \tag{8}$$

by feasibility of $\hat{x}$ for the LP in (1). Finally, we bound the third term $(\langle X, AX \rangle - \langle \hat{x}, A\hat{x} \rangle)$, this time in expectation:

$$\mathbb{E}[\langle X, AX \rangle] - \langle \hat{x}, A\hat{x} \rangle = 0. \tag{9}$$

Chaining eqs. (7) to (9) for the various parts of (6), we get

$$\mathbb{E}[\langle X, AX \rangle] \leq \langle x^*, Ax^* \rangle + (2\varepsilon' + 4\eta)W.$$

Moreover, using that $\langle X, AX \rangle \in [-W, W]$, we get

$$\Pr\left[\langle X, AX \rangle \geq \mathbb{E}[\langle X, AX \rangle] + \eta W\right] = \Pr\left[\langle X, AX \rangle + W \geq \mathbb{E}[\langle X, AX \rangle] + (1 + \eta)W\right]$$
$$\leq \Pr\left[\langle X, AX \rangle + W \geq (1 + \eta/2)\left(\mathbb{E}[\langle X, AX \rangle] + W\right)\right]$$
$$\leq {}^1\!/{(1+\eta/2)}.$$

If $X^*$ is the cut with the smallest value of $\langle X, AX \rangle$ among all the independent roundings:

$$\Pr\left[\langle X^*, AX^* \rangle \leq \langle x^*, Ax^* \rangle + (2\varepsilon' + 5\eta)W\right] \geq 1 - ({}^1\!/{(1+\eta/2)})^T \geq 0.99.$$

Substituting into the definition of $f(\cdot)$ completes the proof. $\qquad\square$

This proves Lemma 3.6, and hence also Theorem 3.4.

*Deterministic Rounding.* We observe that we can replace the repetition by a simple pipeage rounding algorithm to round the fractional solution $\hat{x}$ to an integer solution $X^*$ without suffering any additional loss. Indeed, viewing $\langle x, Ax \rangle$ as a function of some $x_i$ keeping the remaining $\{x_1, \ldots, x_n\} \setminus \{x_i\}$ fixed gives us a linear function of $x_i$ (since the diagonals of $A$ are zero). Hence we can increase or decrease the value of $x_i$ to decrease $\langle x, Ax \rangle$ until $x_i \in \{-1, 1\}$. Iterating over the variables gives the result. However, this does not change the result qualitatively.

## 3.2 Solving MAXCUT for $\Delta$-narrow graphs

Next, we consider $\Delta$-narrow graphs. We show:

**Theorem 3.7.** *For any $\Delta \in \mathbb{N}$, there is a randomized algorithm for the MAXCUT problem with an (expected) approximation factor of $\alpha_{GW} + \tilde{\Omega}(\eta^5/\Delta^2)$ on any $\Delta$-narrow graph.*

For the case of $\Delta$-narrow graphs, we do not use predictions; rather, we adapt an existing algorithm for the MAXCUT problem for low-degree graphs by Feige et al. [2002] and its refinement due to Hsieh and Kothari [2022]. Note that any graph with maximum degree $\Delta$ is clearly $\Delta$-narrow (even when $\eta = 1$).

### 3.2.1 The $\Delta$-narrow Algorithm

We show that Theorem 3.7 holds for the Feige, Karpinski, and Langberg (FKL) MAXCUT algorithm [Feige et al., 2002]. We briefly recall this algorithm first. Consider the MAXCUT SDP with triangle inequalities:

$$\max_{v_i \in S_n \; \forall i \in [n]} \sum_{i < j \in [n]} A_{i,j} \cdot \left(\frac{1 - \langle v_i, v_j \rangle}{2}\right)$$
$$s.t. \quad \|av_i - bv_j\|_2^2 + \|bv_j - cv_k\|_2^2 \geq \|av_i - cv_k\|_2^2 \quad \forall i, j, k \in [n], a, b, c \in \{-1, 1\}.$$

where $S_n$ is the unit sphere of $n$ dimensions. Let $\hat{v}$ be an optimal solution to this SDP.

Let $g$ be a random vector where each coordinate is sampled independently from a standard normal distribution. We use *random hyperplane rounding* from the MAXCUT algorithm of Goemans and Williamson [1995] to round $\hat{v}$ to $\hat{x} \in \{-1, 1\}^n$ as follows: if $\langle \hat{v}_i, g \rangle > 0$, then $\hat{x}_i = 1$; else, $\hat{x}_i = -1$.

Now, define $F = \{i \in [n] : \langle \hat{v}_i, g \rangle \in [-\delta, \delta]\}$ for some $\delta = \Theta(1/((\Delta/\eta)\sqrt{\log(\Delta/\eta)}))$. We partition $N_i := [n]\setminus\{i\}$ as follows: $B_i := \{j \in N_i\setminus F : \hat{x}_j = \hat{x}_i\}$, and $C_i := \{j \in N_i\setminus F : \hat{x}_j \neq \hat{x}_i\}$ and $D_i := N_i \cap F$. We define $F' \subseteq F$ as follows: $i \in F'$ if $i \in F$ and $w(B_i) > w(C_i) + w(D_i)$ where $w(S) := \sum_{j \in S} A_{ij}$. In the final output $X \in \{-1, 1\}^n$, we flip the vertices in $F'$, namely $X_i = -\hat{x}_i$ if $i \in F'$, else $X_i = \hat{x}_i$.

We now give an analysis for the FKL algorithm establishing Theorem 3.7.

The "local gain" for a vertex $i \in F$ is defined as $\Delta_i := (|B_i| - (|D_i| + |C_i|))^+$, where $z^+ = \max(z, 0)$. We now state the following key lemmas:

**Lemma 3.8.** *For any vertex $i \in [n]$, $\Pr[i \in F] = \Omega(\delta)$.*

*Proof.* This lemma immediately follows from [Hsieh and Kothari, 2022, Fact 3]. $\qquad\square$

Let $i$ be a $\Delta$-narrow vertex, and $w \in \mathbb{R}^n$ be its weight vector ($w_i = A_{ij}$ for all $j \in [n]$) so that $W_i = \|w_i\|_1$. Let $w' \in \mathbb{R}^n$ be the projection of $w$ onto its top $\Delta$ coordinates. The narrowness of $i$ implies that $\|w'\|_1 \geq \eta\|w\|_1$, which implies that

$$\|w\|_2^2 \geq \|w'\|_2^2 \geq \frac{\|w'\|_1^2}{\Delta} \geq \frac{\eta^2\|w\|_1^2}{\Delta}.$$

It turns out that the analysis of Hsieh and Kothari [2022] still holds under the above bound between $\ell_1$ and $\ell_2$ norms of weight vectors. So we have the following slight generalization of their Lemma 8.

**Lemma 3.9** (extends Lemma 8 of Hsieh and Kothari [2022])**.** *There is a large enough constant $C$ such that for any $d \geq 3$ and $\delta = \frac{1}{Cd\sqrt{\log d}}$, for any vertex $i$ whose weight vector $w$ satisfies $\|w\|_1^2 \leq d\|w\|_2^2$, it holds that the expected local gain of a vertex $i$ satisfies:*

$$\mathbb{E}[\Delta_i | i \in F] = \Omega\left(\frac{W_i}{d\sqrt{\log d}}\right).$$

*Proof.* In Hsieh and Kothari [2022], the only place where the degree bound $d$ is used is $\|w\|_1^2 \leq d\|w\|_2^2$ at the end of the proof of Lemma 7. $\qquad\square$

*Proof of Theorem 3.7.* Note that the value of the cut $X$ exceeds that of $\hat{x}$ by $\sum_{i \in F'} \Delta_i$, i.e.,

$$\mathbb{E}[\langle X, LX \rangle] = \mathbb{E}[\langle \hat{x}, L\hat{x} \rangle] + \sum_{i \in [n]} \mathbb{E}[\Delta_i | i \in F] \cdot \Pr[i \in F]$$

$$\geq \mathbb{E}[\langle \hat{x}, L\hat{x} \rangle] + \sum_{i:\Delta\text{-narrow}} \mathbb{E}[\Delta_i | i \in F] \cdot \Pr[i \in F].$$

Let the approximation factor of the cut $\hat{x}$ output by the Goemans-Williamson algorithm be denoted $\alpha_{\text{GW}}$ and let opt be the size of the maximum cut. Then,

$$\mathbb{E}[\langle \hat{x}, L\hat{x} \rangle] \geq \alpha_{\text{GW}} \cdot \text{opt}.$$

From Lemmas 3.8 and 3.9 with $d = \Delta/\eta^2$, we get

$$\mathbb{E}[\langle X, LX \rangle] \geq \alpha_{\text{GW}} \cdot \text{opt} + \Omega\left(\frac{1}{(\Delta/\eta^2)\sqrt{\log(\Delta/\eta^2)}} \cdot \sum_{i:\Delta\text{-narrow}} \frac{W_i}{(\Delta/\eta^2)\sqrt{\log(\Delta/\eta^2)}}\right).$$

Since $\sum_{i:\Delta\text{-narrow}} W_i \geq \eta W \geq 2\eta \cdot \text{opt}$, we get

$$\mathbb{E}[\langle X, LX \rangle] \geq (\alpha_{\text{GW}} + \tilde{\Omega}(\eta^5/\Delta^2)) \cdot \text{opt}. \qquad\square$$

### 3.3 Wrapping up: Proof of Theorem 3.1

For $\Delta$-wide graphs, Theorem 3.4 returns a cut with value at least $\text{opt} - (2\eta + \varepsilon')W$ with probability $0.98$. Since we can always find a cut of value $\alpha_{\text{GW}} \cdot \text{opt}$, and $\text{opt} \geq W/2$, this means the expected cut value is at least $\left[0.98 \cdot (1 - 6\eta - 2\varepsilon') + 0.02 \cdot \alpha_{\text{GW}}\right]\text{opt}$. And for $\Delta$-narrow graphs, Theorem 3.7 finds a cut with expected value $\left[\alpha_{\text{GW}} + \tilde{\Omega}(\eta^5/\Delta^2)\right] \cdot \text{opt}$. Moreover, recall that $\Delta = O(1/(\varepsilon\varepsilon')^2)$. Setting $\eta, \varepsilon'$ to be suitably small universal constants gives us that both the above approximation factors are at least $\alpha_{\text{GW}} + \tilde{\Omega}(\varepsilon^4)$, which proves Theorem 3.1.

## 4 MaxCut in the Partial Prediction Model

We now consider the partial prediction model, where each vertex pairwise-independently reveals their correct label with probability $\varepsilon$. Intuitively, this prediction model provides more information than the noisy prediction model since all predictions are guaranteed to be correct. Indeed, this is reflected in out first bound: we show that since an $\varepsilon^2$ fraction of the edges are induced by the vertices with the given labels, it is easy to get an approximation ratio of almost $\alpha_{GW} + \Omega(\varepsilon^2)$. (We give details in Appendix A.)

**Theorem 4.1.** *Given noisy predictions with a rate of $\varepsilon$, there is a polynomial-time randomized algorithm that obtains an (expected) approximation factor of $\alpha_{GW} + \varepsilon^2$ for the* MAXCUT *problem*

Although the $\Omega(\varepsilon^2)$ advantage in this theorem is already better than $\tilde{\Omega}(\varepsilon^4)$ in Theorem 3.1, we ask if can we do even better given the more informative predictions. Ideally, we could get an $\Omega(\varepsilon)$-advantage if the hyperplane rounding performs better than $\alpha_{GW}$ for the edges with only one endpoint's label revealed. One naive way to achieve this is to hope that the rounding *preserves the marginals*; i.e., $\mathbb{E}[x_i] = \langle v_0, v_i \rangle$ for all $i \in [n]$. In that case, if we consider $(i, j)$ where if $v_i = \pm v_0$, the probability that $(i, j)$ is cut is exactly their contribution to the SDP $(1 - \langle v_i, v_j \rangle)$.

Since the hyperplane rounding does not satisfy this property, we use the rounding scheme developed by Raghavendra and Tan [2012] for max-bisection that has an approximation ratio $\alpha_{RT} \approx 0.858$ while preserving the marginals. The proof of this theorem is deferred to Appendix A.

**Theorem 4.2** (Partial Predictions)**.** *Given partial predictions with a rate of $\varepsilon$, there is a polynomial-time randomized algorithm that obtains an (expected) approximation factor of $\alpha_{RT} + (1 - \alpha_{RT} - o(1))(2\varepsilon - \varepsilon^2)$ for the* MAXCUT *problem.*

## 5 2-CSPs in the Noisy Prediction Model

In this section, we go beyond the MAXCUT problem and consider general 2-CSPs. In particular, we extend Theorem 3.4 to this broader class of problems. Let us first define Max-2-CSPs (Constraint Satisfaction Problems). Each constraint is a predicate on two variables: e.g., AND, OR, Not-Equals, or Xor. We are given a collection of such constraints (each on two variables), and the goal is to find an binary assignment to the variables that satisfy the maximum number of constraints. (E.g., if the predicate is Not-Equals, and constraints form the edges of some graph, we get the MAXCUT problem.)

Formally, for a multi-index $\alpha \in [n]^2$ we denote by $\alpha(i)$ its $i$-th index and by $x^\alpha$ the pair $(x_{\alpha(1)}, x_{\alpha(2)})$. For variables $x_1, \ldots, x_n$, we then write $\chi_\alpha(x)$ for the monomial $\prod_{i \in \alpha} x_i$. Given a predicate $P : \{-1, +1\}^2 \to \{0, 1\}$, an instance $\mathcal{I}$ of the CSP(P) problem over variables $x_1, \ldots, x_n$ is a multi-set of triplets $(w, c, \alpha)$ representing constraints of the form $P(c_1 x_{\alpha(1)}, c_2 x_{\alpha(2)}) = 1$ where $\alpha \in [n]^2$ is the scope, $c \in \{\pm 1\}^2$ is the negation pattern and $w \geq 0$ is the weight of the constraint. For brevity we often write $P(c \circ x^\alpha)$ in place of $P(c_1 x_{\alpha(1)}, c_2 x_{\alpha(2)})$. We let $W = \sum_{(w,c,\alpha) \in \mathcal{I}} w$. We can represent the predicate $P$ as the multilinear polynomial of degree 2 in indeterminates $x_{\alpha(1)}, x_{\alpha(2)}$,

$$P(c \circ x^\alpha) = \sum_{\alpha' \subseteq \alpha} c^{\alpha'} \cdot \hat{p}(\alpha') \cdot \chi_{\alpha'}(x),$$

where $\hat{p}(\alpha')$ is the coefficient in $P$ of the monomial $\chi_{\alpha'}(x)$. Notice that this formulation does not rule out predicates with the same multi-index but different negation patterns or multi-indices in which an index appears multiple times. Given a predicate $P$, an instance $\mathcal{I}$ of CSP(P) with $m$ constraints and $x \in \{\pm 1\}^n$, we define

$$\text{val}_{\mathcal{I}}(x) := \frac{1}{W} \sum_{(w,c,\alpha) \in \mathcal{I}} w \cdot P(c \circ x^\alpha) \qquad \text{and} \qquad \text{opt}_{\mathcal{I}} := \max_{x \in \{\pm 1\}^n} \text{val}_{\mathcal{I}}(x).$$

For instance, MAXCUT on a graph $G = ([n], E)$ can be captured in this framework where $P(x, y) = (1 - xy)/2$, each edge $(i, j) \in [n]^2$ yields a triplet $(w, c, \alpha)$ where $w = 1$, $c = (1, 1)$ and $\alpha = (i, j)$.

In the *noisy prediction model* for CSPs, for an instance $\mathcal{I}$ of CSP(P), we assume there is some fixed assigment $x^*$ with value $\text{val}_{\mathcal{I}}(x^*) = \text{opt}_{\mathcal{I}}$. The algorithm has access to a prediction vector

$Y \in \{\pm 1\}^n$ such that predictions $y_i$'s are 2-wise independently correct with probability $\frac{1+\varepsilon}{2}$ for unknown bias $\varepsilon$. We let $Z = \frac{Y}{2\varepsilon}$. With a slight abuse of notation we also write $P(c \circ Z^\alpha)$ even though $Z$ is a rescaled boolean vector.

For a literal $i \in [n]$ and an instance $\mathcal{I}$ of CSP(P) we let $S_i := \{(w, c, \alpha) \in \mathcal{I} \,|\, \alpha(1) = i\}$. As in Section 3.1.1, we can define $\Delta$-wide literals and instances. For an instance $\mathcal{I}$, we partition the constraints in $S_i$ into two sets: the $\Delta$-*prefix* for $i$ comprises the $\Delta$ heaviest constraints in $S_i$ (breaking ties arbitrarily), while the remaining constraints make up the $\Delta$-*suffix* for $i$, which we denote by $\widetilde{S}_i$. We fix a parameter $\eta \in (0, 1/2)$. We let $W_i = \sum_{(w,c,\alpha) \in S_i} w_i$.

**Definition 5.1** ($\Delta$-Narrow/Wide). *A literal $i$ is $\Delta$-wide if the total weight of its in its $\Delta$-prefix is at most $\eta W_i$, and so the weight of edges in its $\Delta$-suffix is at least $(1 - \eta)W_i$. Otherwise, the literal $i$ is $\Delta$-narrow. An instance $\mathcal{I}$ of CSP(P) is $\Delta$-wide if $\sum_{\substack{i \in [n] \\ \Delta\text{-wide}}} W_i \geq (1 - \eta)W$.*

We are now ready to state the main theorem of the section.

**Theorem 5.2.** *Let $P : \{\pm 1\}^2 \to \{0, 1\}$ be a predicate. Let $\varepsilon' \in (0, 1)$, $\eta \in (0, 1/2)$ and $\Delta \geq O(1/(\varepsilon' \cdot \varepsilon)^2)$. There exists a polynomial-time randomized algorithm that, given a $\Delta$-wide $\mathcal{I}$ in CSP(P) and noisy predictions with bias $\varepsilon$, returns a vector $\hat{x} \in \{\pm 1\}^n$ satisfying*

$$\text{val}_{\mathcal{I}}(x) \geq \text{opt}_{\mathcal{I}} - 5\eta - O(\varepsilon'),$$

*with probability at least $0.98$.*

The proof of Theorem 5.2 follows closely that of Theorem 3.4, and is deferred to Appendix B.2.

## 6 Closing Remarks

Our work suggests many directions for future research. One immediate question is to quantitatively improve the exponent of $\varepsilon$ for noisy predictions, and the constants. Here are some broader questions:

1. We assume that our noisy predictions are correct with probability *equal to* $1/2 + \varepsilon$; we can easily extend to the case where each node has a prediction that is correct with some probability $1/2 + \varepsilon_i$, and each $\varepsilon_i \in \Theta(\varepsilon)$. Can we extend to the case when we are only guaranteed $\varepsilon_i \geq \varepsilon$ for every $i$?

2. For which other problems can we improve the performance of the state-of-the-art algorithms using noisy predictions? As we showed, the ideas used for the $\Delta$-wide case extend to more general maximization problems on 2-CSPs with "high degree", but can we extend the results for the "low-degree" case where each variable does not have a high-enough degree to infer a clear signal? Can we extend these to minimization versions of 2-CSPs?

3. What general lower bounds can we give for our prediction models? We feel that $\alpha_{GW} + O(\varepsilon)$ is a natural barrier. One "evidence" we have is the following integrality gap for the SDP used in the partial information model; starting from a gap instance and an SDP solution exhibiting $\text{opt} \leq \alpha_{GW} \cdot \text{sdp}$ for the standard SDP (without incorporating revealed information), given labels for an $\varepsilon n$ vertices, our new SDP simply fixes the positions of the corresponding $\varepsilon n$ vectors, but doing that from the given SDP solution decreases the SDP value by at most $O(\varepsilon)$ in expectation, which still yields $\text{opt} \leq (\alpha_{GW} + O(\varepsilon))\text{sdp}$. (Note that you can replace the SDP gap with any hypothetical gap instance for stronger relaxations.)

   Given that the partial predictions model is easier than the noisy predictions model and our entire algorithm for the partial model is based on this SDP, this might be considered as a convincing lower bound, but it would be nicer to have more general lower bounds against all polynomial-time algorithms.

4. Our models only assume pairwise independence between vertices: can we extend our results to other ways of modeling correlations between the predictions? In addition to stochastic predictions, can we incorporate geometric predictions (e.g., in random graph models where the probability of edges depend on the proximity of the nodes)?

## Acknowledgments

We thank Ola Svensson for enjoyable discussions. AG was supported in part by NSF awards CCF-2006953 and CCF-2224718, and by Google, Inc. EL was supported in part by NSF award CCF-2236669 and by Google, Inc. DP was supported in part by NSF awards CCF-1955703 and CCF-2329230, and by Google, Inc.

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

# Appendix

## A  Missing proofs for MAXCUT in the partial prediction model

*Proof of Theorem 4.1.* Given a graph $G = (V, E)$ with the optimal cut $(A^*, B^*)$ that cuts $E^* = E \cap E(A^*, B^*)$, let $S$ be the set of vertices whose label is given, and let $A = A^* \cup S$, $B = B^* \cup S$. Consider the following MAXCUT SDP that fixes the vertices with the revealed labels.

$$\max_{v_i \in S_n \ \forall i \in [n]} \sum_{i,j \in [n]} \frac{A_{i,j}(1 - \langle v_i, v_j \rangle)}{2} \quad s.t. \ v_i = v_0 \ \forall i \in A \text{ and } v_i = -v_0 \ \forall i \in B.$$

Note that this is still a valid relaxation so the optimal SDP value sdp is at least opt. For each edge $e \in E^*$, $e \in E(A, B)$ with probability $\varepsilon^2$; in other words, both of its endpoints will reveal their labels. Let $\tau$ denotes the total weight of such edges, so that $\mathbb{E}[\tau] = \varepsilon^2 \text{opt}$. Note that $\text{sdp} \geq \text{opt}$ for every partial prediction.

Perform the standard hyperplane rounding. For each $e \in E^* \cap E(A, B)$, the rounding will always cut $e$. For all other edges, we have an approximation ratio of $\alpha_{\text{GW}}$. Therefore, the expected weight of the cut edges is at least

$$\mathbb{E}[\tau W + \alpha_{\text{GW}}(\text{sdp} - \tau W)] \geq \varepsilon^2 \text{opt} + \alpha_{\text{GW}}(1 - \varepsilon^2)\text{opt} = (\alpha_{\text{GW}} + (1 - \alpha_{\text{GW}})\varepsilon^2) \cdot \text{opt}. \quad \square$$

*Proof of Theorem 4.2.* Given a graph $G = (V, E)$ with the optimal cut $(A^*, B^*)$ that cuts $E^* = E \cap E(A^*, B^*)$, let $S$ be the set of vertices whose label is given, and let $A = A^* \cup S$, $B = B^* \cup S$. Let $E'$ be the set of the edges that are incident on $A$ or $B$. Each edge cut in the optimal solution will be in $E'$ with probability $2\varepsilon - \varepsilon^2$. Let $\tau$ be the total weight of the edges in $E^* \cap E'$ so that $\mathbb{E}[\tau] = (2\varepsilon - \varepsilon^2)\text{opt}$. Guess the value of $\tau$ (up to a $o(1)$ multiplicative error that we will ignore in the proof), and consider the following MAXCUT SDP that fixes the vertices with the revealed labels and requires a large SDP contribution from $E'$.

$$\max_{v_i \in S_n \ \forall i \in [n]} \sum_{i,j \in [n]} \frac{A_{i,j}(1 - \langle v_i, v_j \rangle)}{2}$$

$$s.t. \ v_i = v_0 \qquad\qquad\qquad\qquad \forall i \in A$$

$$v_i = -v_0 \qquad\qquad\qquad\qquad \forall i \in B$$

$$\sum_{(i,j) \in E'} \frac{A_{i,j}(1 - \langle v_i, v_j \rangle)}{2} \geq \tau.$$

Given the correctly guessed value of $\tau$, the optimal solution is still feasible for the above SDP, so $\text{sdp} \geq \text{opt}$. We use Raghavendra and Tan [2012]'s rounding algorithm, which is briefly recalled below.

- For each $i \in [n]$, define $\mu_i \in [-1, +1]$ and $w_i \in \mathbb{R}^n$ such that $v_i = \mu_i v_0 + w_i$ and $w_i \perp v_0$. Let $\overline{w_i} = w_i / \|w_i\|$. ($w_i = 0$ if and only if $v_i = \pm v_0$. Then define $\overline{w_i} = 0$.)

- Pick a random Gaussian vector $g$ orthogonal to $v_0$. Let $\xi_i := \langle g, \overline{w_i} \rangle$. Note that each $\xi_i$ is a standard Gaussian.

- Let the threshold $t_i := \Phi^{-1}(\mu_i/2 + 1/2)$ where $\Phi$ is the CDF of a standard Gaussian.

- If $\xi_i \leq t_i$, set $x_i = 1$ and otherwise set $x_i = -1$.

Raghavendra and Tan showed that this rounding achieves an $(\alpha_{\text{RT}} \approx 0.858)$-approximation for every edge.

Consider an edge $(i, j) \in E'$ and without loss of generality, assume $i \in B$, which implies that $v_i = -v_0$. The contribution of this edge to the SDP objective is $\mu_j/2 + 1/2$. Note that $\Pr[x_j = 1]$ is exactly $\mu_j/2 + 1/2$ and $\mathbb{E}[x_j] = (\mu_j/2 + 1/2) - (1/2 - \mu_j/2) = \mu_j$. So, we get a 1-approximation from this edge. Since other edges still have an $\alpha_{\text{RT}}$-approximation, the total expected weight of the edges cut is at least

$$\mathbb{E}[\tau + \alpha_{\text{RT}}(\text{sdp} - \tau)] \geq (2\varepsilon - \varepsilon^2)\text{opt} + \alpha_{\text{RT}}(1 - (2\varepsilon - \varepsilon^2))\text{opt} = \alpha_{\text{RT}} \cdot \text{opt} + (1 - \alpha_{\text{RT}})(2\varepsilon - \varepsilon^2)\text{opt}.$$

Hence the proof of Theorem 4.2. $\qquad\qquad\qquad\qquad\qquad\qquad\qquad\qquad\qquad\qquad\qquad\qquad\qquad \square$

# B  Missing details for 2-CSPs

The goal of this section is to establish Theorem 5.2 for dense instances of 2-CSPs.

## B.1  The $2$-CSP Algorithm for Dense Instances

First observe that we may assume without loss of generality that each $(w, c, \alpha)$ appears exactly twice in $\mathcal{I}$. This is convenient so that for all $x \in \{\pm 1\}^n$, $\mathrm{val}_{\mathcal{I}}(x) = \sum_{i \in [n]} \sum_{(w,c,\alpha) \in S_i} w \cdot P(c \circ x^{\alpha})$. With a slight abuse of notation, for all $(w, c, \alpha) \in S_i$, we let

$$P(c \circ (x_i \cdot Z^{\alpha \backslash i})) := \sum_{\substack{\alpha' \subseteq \alpha \text{ s.t.} \\ \alpha'(1)=i}} \hat{p}_{\alpha'} c^{\alpha'} x_i \cdot \chi_{\alpha' \backslash \alpha'(1)}(Z) + \sum_{\substack{\alpha' \subseteq \alpha \text{ s.t.} \\ \alpha'(1) \neq i}} \hat{p}_{\alpha'} c^{\alpha'} \chi_{\alpha'}(Z),$$

and

$$P(c \circ x^{\alpha \backslash i}) := \sum_{\substack{\alpha' \subseteq \alpha \text{ s.t.} \\ \alpha'(1)=i}} \hat{p}_{\alpha'} c^{\alpha'} \chi_{\alpha' \backslash \alpha'(1)}(x) + \sum_{\substack{\alpha' \subseteq \alpha \text{ s.t.} \\ \alpha'(1) \neq i}} \hat{p}_{\alpha'} c^{\alpha'} \chi_{\alpha'}(x),$$

We further define $\tilde{S}_i \subseteq S_i$ to be subset of constraints in $S_i$ that are not part of the $\Delta$-prefix of $i$.

We can now state the algorithm behind Theorem 5.2, which amounts to the following two steps.

1. Solve the linear program

$$\max_{x \in [-1,+1]^n} \sum_{\substack{i \in [n] \\ \Delta\text{-wide}}} \sum_{(w,c,\alpha) \in \tilde{S}_i} w P(c \circ (x_i \cdot Z^{\alpha \backslash i}))$$

   subject to

$$\sum_{\substack{i \in [n] \\ \Delta\text{-narrow}}} \left| \sum_{(w,c,\alpha) \in S_i} w P(c \circ x^{\alpha \backslash i}) \right| + \sum_{\substack{i \in [n] \\ \Delta\text{-wide}}} \left| \sum_{(w,c,\alpha) \in S_i \backslash \tilde{S}_i} w P(c \circ x^{\alpha \backslash i}) \right|$$

$$+ \sum_{\substack{i \in [n] \\ \Delta\text{-wide}}} \left| \sum_{(w,c,\alpha) \in \tilde{S}_i} w \left( P(c \circ x^{\alpha \backslash i}) - P(c \circ Z^{\alpha \backslash i}) \right) \right| \leq C(\varepsilon' + 2\eta)W \qquad (10)$$

   for some large enough absolute constant $C > 0$. Let $\hat{x} \in [-1, +1]^n$ be the found optimal solution.

2. Repeat $O(1/\eta)$ times independently and output the best assignment $X^*$: independently for each $i \in [n]$ set $X_i = 1$ with probability $(1 + \hat{x}_i)/2$ and $X_i = -1$ otherwise.

The LP above generalize the one in eq. (1), which comes as a special case where $\hat{p}_{\alpha'} = 0$ for all $\alpha' \subset \alpha \in [n]^2$. Indeed, since predicates contain only two literals, the program is linear.

## B.2  The Analysis of the $2$-CSP Algorithm

We obtain here the proof of Theorem 5.2.

**Feasibility of the best assignment**  As in Lemma 3.5, we first prove that, in expectation over the prediction $Y$, $x^*$ is a feasible solution to the program.

**Lemma B.1.** *Consider the settings of Theorem 5.2. Then*

$$\mathbb{E} \sum_{\substack{i \in [n] \\ \Delta\text{-narrow}}} \left| \sum_{(w,c,\alpha) \in S_i} w P(c \circ x^{*\alpha \backslash i}) \right| + \sum_{\substack{i \in [n] \\ \Delta\text{-wide}}} \left| \sum_{(w,c,\alpha) \in S_i \backslash \tilde{S}_i} w P(c \circ x^{*\alpha \backslash i}) \right|$$

$$+ \sum_{\substack{i \in [n] \\ \Delta\text{-wide}}} \left| \sum_{(w,c,\alpha) \in \tilde{S}_i} w \left( P(c \circ x^{*\alpha \backslash i}) - P(c \circ Z^{\alpha \backslash i}) \right) \right| \leq W(2\eta + O(1/\varepsilon\sqrt{\Delta})).$$

*Proof.* First, by definition of $\Delta$-wide instance,

$$\sum_{\substack{i \in [n] \\ \Delta\text{-narrow}}} \left| \sum_{(w,c,\alpha) \in S_i} wP(c \circ x^{*\alpha \setminus i}) \right| \le \eta W.$$

Second, by definition for any $\Delta$-wide vertex $i$,

$$\left| \sum_{(w,c,\alpha) \in S_i \setminus \tilde{S}_i} wP(c \circ x^{*\alpha \setminus i}) \right| \le \eta W_i.$$

Hence it remains to show

$$\mathbb{E} \sum_{\substack{i \in [n] \\ \Delta\text{-wide}}} \left| \sum_{(w,c,\alpha) \in \tilde{S}_i} w\left( P(c \circ x^{*\alpha \setminus i}) - P(c \circ Z^{\alpha \setminus i}) \right) \right| \le O(W/(\varepsilon \sqrt{\Delta})).$$

Now, recall that $\mathbb{E}[Y_i] = 2\varepsilon x_i^*$ and thus $\mathbb{E}[Z] = x^*$. So for any $(c, \alpha) \in \mathcal{I}$, $\mathbb{E}[P(c \circ Z^\alpha)] = \mathbb{E}[P(c \circ x^{*\alpha})]$ by pair-wise independence of the predictions. Thus it suffices to study, for each $\Delta$-wide $i$, $\mathrm{var}\left( \sum_{(w,c,\alpha) \in S_i} wP(c \circ Z^{\alpha \setminus i}) \right)$. To this end, notice that for any $\alpha, \alpha' \in S_i$ with $\alpha \cap \alpha' = \{i\}$ it holds

$$\mathbb{E}\left[ P(c \circ Y^{\alpha \setminus i}) P(c \circ Y^{\alpha' \setminus i}) \right] = \mathbb{E}\left[ P(c \circ Y^{\alpha \setminus i}) \right] \mathbb{E}\left[ P(c \circ Y^{\alpha' \setminus i}) \right].$$

Moreover, since $|\alpha| = 2$, there are at most 4 distinct negation patterns. Therefore, by the AM-GM inequality

$$\mathrm{var}\left( \sum_{(w,c,\alpha) \in \tilde{S}_i} wP(c \circ Z^{\alpha \setminus i}) \right) \le \sum_{(w,c,\alpha) \in \tilde{S}_i} O(w^2) \, \mathrm{var}\left( P(c \circ Z^{\alpha \setminus i}) \right)$$

$$\le \sum_{(w,c,\alpha) \in \tilde{S}_i} O\left( \frac{w^2}{\varepsilon^2} \right)$$

where we used the fact that entries of $Z$ are bounded by $1/\varepsilon$ and the coefficients of a boolean predicate are bounded by 1 (by Parseval's Theorem, see O'Donnell [2014]). By construction of $\tilde{S}_i$, each $(w, c, \alpha) \in \tilde{S}_i$ must satisfy $w \le W_i / \Delta$. Using Holder's inequality

$$\mathrm{var}\left( \sum_{(w,c,\alpha) \in \tilde{S}_i} wP(c \circ Z^{\alpha \setminus i}) \right) \le O\left( \frac{W_i^2}{\Delta \cdot \varepsilon^2} \right).$$

We can use this bound on the variance in combination with Chebishev's inequality to obtain, for $\lambda > 0$,

$$\mathbb{P}\left( \left| \sum_{(w,c,\alpha) \in S_i} w\left( P(c \circ x^{*\alpha \setminus i}) - P(c \circ Z^{\alpha \setminus i}) \right) \right| \ge \lambda \right) \le O\left( \frac{W_i^2}{\varepsilon^2 \cdot \Delta \cdot \lambda^2} \right).$$

Let $\lambda := O(W_i / (\varepsilon \sqrt{\Delta}))$. A peeling argument now completes the proof:

$$\mathbb{E}\left[ \left| \sum_{(w,c,\alpha) \in S_i} w\left( P(c \circ x^{*\alpha \setminus i}) - P(c \circ Z^{\alpha \setminus i}) \right) \right| \right]$$

$$\le \lambda + \sum_{t \ge 0} 2^{t+1} \lambda \cdot \mathbb{P}\left( \left| \sum_{(w,c,\alpha) \in S_i} w\left( P(c \circ x^{*\alpha \setminus i}) - P(c \circ Z^{\alpha \setminus i}) \right) \right| \ge 2^t \lambda \right) \le O(\lambda).$$

$\square$

**Analysis of the algorithm**  We can use Lemma B.1 to obtain our main theorem for CSPs.

*Proof of Theorem 5.2.*  We follow closely the proof of Lemma 3.6. Consider one of the assignments $X \in \{\pm 1\}^n$ found in the second step of the algorithm. Recall $\hat{x} \in [-1, +1]^n$ denotes the optimal fractional solution found by the algorithm. We may rewrite for each $\Delta$-wide $i$ and $(w, c, \alpha) \in \tilde{S}_i$

$$
\sum_{\substack{i \in [n] \\ \Delta\text{-wide}}} \sum_{(w,c,\alpha) \in \tilde{S}_i} wP(c \circ X^\alpha) = \sum_{\substack{i \in [n] \\ \Delta\text{-wide}}} \sum_{(w,c,\alpha) \in \tilde{S}_i} w \Big[ P(c \circ (\hat{x}_i \cdot Z^{\alpha \backslash i}))
$$
$$
+ P(c \circ X^\alpha) - P(c \circ \hat{x}^\alpha)
$$
$$
+ P(c \circ \hat{x}^\alpha) - P(c \circ (\hat{x}_i \cdot Z^{\alpha \backslash i})) \Big] . \tag{11}
$$

We bound each term in Equation (11) separately. First, notice that by Markov's inequality and Lemma B.1, with probability $0.99$, $x^*$ is a feasible solution to the LP. Conditioning on this event $\mathcal{E}$

$$
\sum_{\substack{i \in [n] \\ \Delta\text{-wide}}} \sum_{(w,c,\alpha) \in \tilde{S}_i} wP(c \circ (\hat{x}_i \cdot Z^{\alpha \backslash i})) \geq \sum_{\substack{i \in [n] \\ \Delta\text{-wide}}} \sum_{(w,c,\alpha) \in \tilde{S}_i} wP(c \circ (x_i^* \cdot Z^{\alpha \backslash i}))
$$
$$
= \sum_{\substack{i \in [n] \\ \Delta\text{-wide}}} \sum_{(w,c,\alpha) \in \tilde{S}_i} wP(c \circ x^{*\alpha})
$$
$$
+ \sum_{\substack{i \in [n] \\ \Delta\text{-wide}}} \sum_{(w,c,\alpha) \in \tilde{S}_i} w \Big( P(c \circ (x_i^* \cdot Z^{\alpha \backslash i})) - P(c \circ x^{*\alpha}) \Big)
$$

By Holder's inequality and the fact that $x^*$ is feasible, for $\Delta$-wide $i$,

$$
\sum_{\substack{i \in [n] \\ \Delta\text{-wide}}} \sum_{(w,c,\alpha) \in \tilde{S}_i} w \Big( P(c \circ (x_i^* \cdot Z^{\alpha \backslash i})) - P(c \circ x^{*\alpha}) \Big)
$$
$$
\leq \sum_{\substack{i \in [n] \\ \Delta\text{-wide}}} \left| \sum_{(w,c,\alpha) \in \tilde{S}_i} w \Big( P(c \circ Z^{\alpha \backslash i}) - P(c \circ x^{*\alpha \backslash i}) \Big) \right| \leq (O(\varepsilon') + 2\eta)W .
$$

Since by construction $\hat{x}$ is feasible, another application of Holder's inequality also yields the following bound on the third term,

$$
\sum_{\substack{i \in [n] \\ \Delta\text{-wide}}} \sum_{(w,c,\alpha) \in \tilde{S}_i} w \Big( P(c \circ (\hat{x}_i \cdot Z^{\alpha \backslash i})) - P(c \circ \hat{x}^\alpha) \Big) \leq (O(\varepsilon') + 2\eta)W .
$$

For the second term in Equation (11), by construction of $X$ we have $\mathbb{E}\left[ P(c \circ X^\alpha) \,|\, \mathcal{E} \right] = P(c \circ \hat{x}^\alpha)$. Combining the three bounds, we get that

$$
\mathrm{opt}_\mathcal{I} \geq E \left[ \frac{1}{W} \sum_{\substack{i \in [n] \\ \Delta\text{-wide}}} \sum_{(w,c,\alpha) \in \tilde{S}_i} wP(c \circ X^\alpha) \,\middle|\, \mathcal{E} \right] \geq \mathrm{opt}_\mathcal{I} - (O(\varepsilon') + 4\eta) .
$$

Applying Markov's inequality on the random variable $\mathrm{opt}_\mathcal{I} - \frac{1}{W} \sum_{\substack{i \in [n] \\ \Delta\text{-wide}}} \sum_{(w,c,\alpha) \in \tilde{S}_i} wP(c \circ X^\alpha)$, we get

$$
\mathbb{P}\left( \frac{1}{W} \sum_{\substack{i \in [n] \\ \Delta\text{-wide}}} \sum_{(w,c,\alpha) \in \tilde{S}_i} wP(c \circ X^\alpha) \leq \mathrm{opt}_\mathcal{I} - (O(\varepsilon') + 5\eta) \,\middle|\, \mathcal{E} \right) \leq \frac{1}{1+\eta}
$$

The theorem follows since we sample $O(1/\eta)$ independent assignments $X$ and pick the best.  $\square$

