# OpenReview forum: "Learning-Augmented Approximation Algorithms for Maximum Cut and Related Problems"
_NeurIPS.cc/2024/Conference — NeurIPS 2024 poster_

### Official Review · Reviewer_fPP4 · 2024-07-09

**Soundness:** 3
**Presentation:** 3
**Contribution:** 4
**Rating:** 7
**Confidence:** 3

**Summary:**

For the Max Cut Problem, the authors first consider predictions that are independently correct with probability 1/2 + $\epsilon$ (noisy prediction model). They obtain an approximation guarantee of $0.878 + \Omega(\epsilon^4)$, improving upon best-known approximation guarantees. The idea of the algorithm is to divide instances by wide and narrow graphs, a notion related to high-degree and low-degree graphs. By obtaining approximation guarantees for both cases and optimizing over some parameters, they obtain their first main result.
In the second setting, they consider a partial prediction model, in which one gets a correct prediction, but only for an $\epsilon$-fraction of randomly chosen vertices. In this case they obtain a $0.858 + \Omega(\epsilon)$ approximation algorithm, which is slightly worse than the $0.878$-approximation for small $\epsilon$, but in general the dependency on $\epsilon$ is more favorable. The slightly worse constant is because one needs to employ a rounding scheme from Raghavendra and Tan, which has this approximation guarantee. However, by doing this, one obtains a better dependency on $\epsilon$.
Finally, the authors also consider 2-CSPs in the noisy prediction model. The ideas are similar to the case for the Max Cut Problem.

**Strengths:**

The paper is well-written and the proofs seem sound. The results are nice and improve upon state-of-the-art approximation algorithms given good predictions on the solution. To the best of my knowledge, this paper is the first to substantially contribute towards learning-augmented algorithms in the area of approximation algorithms. Hence, the paper is innovative and might lead to new results in this interesting area.

**Weaknesses:**

The results are specifically tailored for the problems considered in this paper. However, this is no surprise as this paper is the first to study learning-augmented algorithms in this area. Therefore, I do not see this as a strong weakness.

**Questions:**

-

**Limitations:**

In my opinion, all limitations have been properly addressed.

---

> ### Author Response · Authors · 2024-08-05
>
> We thank the reviewer for the comments and the constructive encouragement.

---

### Official Review · Reviewer_969c · 2024-07-12

**Soundness:** 3
**Presentation:** 3
**Contribution:** 3
**Rating:** 7
**Confidence:** 4

**Summary:**

Authors study algorithms constraint satisfaction problems, in particular, MAX-CUT
which are provided with predictions mildly correlated with the optimal solution.
In the case of MAX-CUT, we have a prediction +1 or -1 for each vertex of the graph
which suggests which side of the maximum cut should it belong to.
For each vertex, this prediction is correct with probability 0.5+eps,
assuming a pairwise independence between the predictions.
They call this eps-correlated predictions.
They also consider a similar model where all predictions are guaranteed to be correct
but they are available only for eps-fraction of the vertices chosen randomly.

Their main result is a poly-time algorithm for MAX-CUT which, given
eps-correlated predictions, achieves an improvement of order Omega(eps^4)
over the best possible approximation ratio achievable by a poly-time algorithm
without any predictions.
Their algorithm is based on combination of linear programming and semidefinite
programming techniques.

**Strengths:**

* I believe that the problem of solving MAX-CUT
starting with a solution which is mildly correlated with the optimum is a basic
question about SDP techniques for MAX-CUT. Since MAX-CUT is one of the central
problems in theory of algorithms, I consider the progress on this question to
be the main strength of this paper.

* the main clear model of MAX-CUT with eps-correlated predictions is also extended
to CSPs and also to the model with partial predictions.

**Weaknesses:**

Nothing particular.

**Questions:**

* Is the precision parameter of the predictions epsilon known to the algorithm?

* Many works, especially for online problems, consider predictions whose error
is distributed adversarially. Do you think such predictions are also applicable
to MAX-CUT and CSPs?

**Limitations:**

properly stated in the theorem statements.

---

> ### Author Response · Authors · 2024-08-05
>
> We thank the reviewer for the comments and the constructive feedback.
>
> - *Is the precision parameter of the predictions epsilon known to the algorithm?*
>
> The parameter $\varepsilon$ does not need to be known: for example, we can run our algorithm for each $\varepsilon$ that’s a power of $\frac{1}{2}$ (between $\frac{1}{n}$ and $1$), and return the best of these solutions. One of these will be the right choice of $\varepsilon$, and our analysis holds for that choice of $\varepsilon$. (We will clarify this in the next version of the paper.)
>
> - *Many works, especially for online problems, consider predictions whose error is distributed adversarially. Do you think such predictions are also applicable to MAX-CUT and CSPs?*
>
> In general, giving the adversary power to choose the vertices it gives correct information on can make the predictions useless. For example, consider an instance containing $(1-\delta)n$ isolated vertices, and an arbitrary “hard” graph on the remaining $\delta n$ vertices. Now the adversary may only provide correct predictions on those isolated vertices.
>
> So, we need to make some assumptions to avoid these pathologies. For example, if we assume that the graph is regular, our Theorem 4.1 for partial predictions that gives $0.858 + \Omega(\epsilon)$ approximation goes through even with deterministic predictions. A similar result holds in general graphs for deterministic predictions satisfying other properties. E.g., if an $\epsilon$ fraction of neighbors’ labels are revealed for every vertex, or more generally, if an $\epsilon$ fraction of edges are incident to the vertices whose labels are revealed. We will include a discussion about deterministic predictions in the Closing Remarks. We hope that our work will spur future investigation into such prediction models.

---

> > ### Comment · Reviewer_969c · 2024-08-09
> >
> > thank you for your answers.

---

### Official Review · Reviewer_SRjy · 2024-07-12

**Soundness:** 4
**Presentation:** 4
**Contribution:** 4
**Rating:** 7
**Confidence:** 3

**Summary:**

**Problem Studied**

This paper studies the Max Cut and 2-CSP problems in a setting where there is some noisy prediction of the optimal solution. In particular, the paper considers the following three settings:
- Max Cut in the "noisy prediction model": Here, each vertex gives its true label with probability $\frac{1}{2} + \epsilon$ (and its opposite label otherwise).
- Max Cut in the "partial prediction model": Here, an $\epsilon$-fraction of the vertices reveal their true label at random.
- 2-CSPs in the noisy prediction model.

**Main Results / Contributions**

For Max Cut, the main result is that it is possible to beat the Goemans-Williamson approximation factor when $\epsilon > 0$. For the noisy prediction model, the authors obtain an approximation ratio that is roughly $\alpha_{GW} +  \Omega(\epsilon^4)$. For the partial prediction model, it is possible to get $\alpha_{GW} + \Omega(\epsilon^2)$, or $\alpha_{GT} + \Omega(\epsilon)$, where $\alpha_{GT} \approx 0.858$ is the approximation ratio of the algorithm by Raghavendra and Tan.

The authors extend one result to 2-CSPs in the noisy prediction model that are "wide".

**Strengths:**

The paper is very well written. It is nice to read and the contributions of the paper are clear. It studies a natural variant (prediction-augmented) of a well-known problem (max cut). It also answers an open question posed by Ola Svensson, which is a good sign.

**Weaknesses:**

The title is perhaps promising more than the paper delivers -- the paper only studies Max Cut and a subclass of 2-CSP instances, so it is a bit misleading to use "maximization problems" in the title.

**Questions:**

1. For the noisy prediction model of Max Cut, a natural algorithm that comes to mind is to return the better of the Goemans-Williamson cut and the predicted cut. Is there an example where this algorithm does not do better than $\alpha_{GW}$?

2. In the literature on algorithms with predictions, one common way people have modeled the "prediction" is as a single predicted solution, instead of assuming it is generated from a distribution. They then usually parameterize the performance of the algorithm (e.g. approximation ratio), as a function of the "error" in the (single) predicted solution, where "error" is something that has to be defined depending on the problem. This has the advantage of not needing to define the form of the distribution that the prediction comes from (like the noisy prediction model or the partial prediction model), and gives a somewhat more "instance-dependent" guarantee. What are your thoughts on this way of viewing the prediction in the context of your problem? Do you think it is a better or worse model, and why?

3. As $\epsilon \to \frac{1}{2}$ in the noisy predictions model, the prediction approaches the optimal solution, and so the approximation ratio of the algorithm should intuitively go to 1. Is this the case for the algorithm considered in this paper? This is not clear from the theorem statement because it has a big-Omega in the bound. Is there is a way to write the approximation ratio exactly in terms of $\epsilon$ to show that it approaches 1 as $\epsilon \to \frac{1}{2}$? If so, I think that would be instructive. Similarly for the partial predictions model as $\epsilon \to 1$.

4. What is the intuition for why predictions are not needed for narrow instances?

**Limitations:**

Yes.

---

> ### Author Response · Authors · 2024-08-05
>
> Thanks for the comments and constructive encouragement.
>
> - *For the noisy prediction model of Max Cut, a natural algorithm that comes to mind is to return the better of the GW cut and the predicted cut. Is there an example where this algorithm does not do better than $\alpha_{GW}$?*
>
> Yes: consider the tight instances for GW rounding (ones that achieve $\alpha_{GW}$ times the max-cut): such instances were given by Alon, Sudakov, and Zwick, and have maximum cuts that cut about 84.5% of the edges. Given any graph, the predicted labeling cuts each edge in the max-cut with probability $\frac{1}{2} + 2 \varepsilon^2$ and every other edge with probability $\frac{1}{2} - 2 \varepsilon^2$, thereby cutting a bit more than half the edges of the graph for small $\varepsilon$. So on the tight instance graphs, GW rounding would give an $\alpha_{GW}$ approximation (by design), and the predicted cut would give $\approx 0.5/(0.845) \ll \alpha_{GW}$ approximation.
>
> - *In the literature on algorithms with predictions, one common way people have modeled the "prediction" is as a single predicted solution, instead of assuming it is generated from a distribution.[..] What are your thoughts on this way of viewing the prediction in the context of your problem?*
>
> Such a model would certainly make sense, but the qualitative bounds would depend on the instances. In particular, without additional assumptions, giving the adversary power to choose the partition can make the predictions useless despite being correct on almost all the vertices. For example, consider an instance containing $(1-\delta)n$ isolated vertices, and an arbitrary “hard” graph on the remaining $\delta n$ vertices. Now the adversary may only provide correct predictions on those isolated vertices.
>
> So, we need to make some assumptions to avoid these pathologies. For example, if we assume that the graph is regular, our Theorem 4.1 for partial predictions that gives $0.858 + \Omega(\epsilon)$ approximation goes through even with deterministic predictions. A similar result holds in general graphs for deterministic predictions satisfying other properties. E.g., if an $\epsilon$ fraction of neighbors’ labels are revealed for every vertex, or more generally, if an $\epsilon$ fraction of edges are incident to the vertices whose labels are revealed. We will include a discussion about deterministic predictions in the Closing Remarks. We hope that our work will spur future investigation into such prediction models.
>
> - *As 𝜖→1/2 in the noisy predictions model, the prediction approaches the optimal solution, and so the approximation ratio of the algorithm should intuitively go to 1. Is this the case for the algorithm considered in this paper? […]*
>
> Our algorithms indeed achieve a near-perfect approximation as the advice becomes nearly perfect. For the noisy model, when $\epsilon=1/2 - \delta$ for small $\delta>0$, it can be shown that a simplification of our algorithm in Section 3.1 guarantees an $(1-O(\sqrt(\delta)))$-approximation. The simplification is just ignoring $\Delta$ and $\eta$, wideness and prefixes, and letting $\tilde{A} = A$. Then Lemma 3.5 can be shown to hold with the the right hand side replaced by $O(\sqrt{\delta} W)$, where the crucial change in the proof is the upper bound on $Var(Z_j)$ in line 220 from $O(1/\epsilon^2)$ to $O(\delta)$.
>
> For the partial prediction model, when $\epsilon = 1-\delta$ for small $\delta>0$, Theorem 4.2’s guarantee $\alpha_{RT} + (1 - \alpha_{RT} - o(1))(2\epsilon - \epsilon^2)$ becomes $1 - O(\delta^2)$.
>
> We will try to mention the above results in the final version of the paper.
>
> - *What is the intuition for why predictions are not needed for narrow instances?*
>
> For this discussion, consider unweighted graphs. Algorithms improving on GW for low-degree graphs use two ideas: a stronger SDP relaxation using triangle inequalities, and a local search after rounding. The GW SDP has a tight integrality gap even on degree-2 instances, so using the stronger SDP is unavoidable.
> *Why does local search help on low-degree graphs?* Well, for high-degree vertices, tail concentration typically forces each vertex to have more neighbors on the other side of the cut after rounding. But this is not the case for low-degree graphs because concentration bounds are weaker, which means that locally improving moves is a viable strategy for improving the rounded solution with non-trivial probability. For a longer discussion, please see the paper of Feige, Karpinski, and Langberg that gave the first algorithm to beat the GW bound on low-degree graphs.
>
> For weighted graphs, all this intuition carries over, but we need to be careful how the weight of the edges incident to a vertex are distributed: whether it is concentrated on a few edges (“narrow”) or is spread over many edges (“wide”). (We will add the intuition in the next version of the paper.)

---

> > ### Comment · Reviewer_SRjy · 2024-08-12
> >
> > Thank you for your detailed response, I appreciate it!

---

### Official Review · Reviewer_PkHv · 2024-07-13

**Soundness:** 3
**Presentation:** 3
**Contribution:** 3
**Rating:** 5
**Confidence:** 3

**Summary:**

The authors discuss about a setup if the approximation ratio of the known approximation algorithms for offline NP-hard problems can be improved in the cases where we have access to noisy or partial predictions. They answer this investigation positively for MaxCut and Constraint Satisfaction (CSP) problems. Their theoretical investigation proves that the worst-case performance for the MaxCut and 2-CSP problems can be improved with assuming access to pairwise independent predictions that are correct with the probability $1/2 + \epsilon$. This corresponds to the noisy predictions setting. In the partial prediction settings, they assume that each vertex reveals their correct label with probability $\epsilon$ pairwise independently. Similarly, this theoretically improves the worst-case performance for the MaxCut problem.

**Strengths:**

The central motivation of the paper is quite interesting. With the abundance of the prediction ML models, utilizing their outputs in improving the approximation guarantees of famous combinatorial optimization problems makes it a challenging discussion. The problem setups are explained clearly. Considering much of the related work focuses on the online setting, making contributions for the offline setting seems significant.

**Weaknesses:**

- The introduction could motivate the readers more about the significance of the offline setting.
- On the first glance, it is not very clear how this papers differs from the concurrent work mentioned on the last paragraph of the Related Work (besides one of them making the full independence assumption between the predictions).
- The existence of predictions is treated as a given but often times acquiring those predictions comes with a significant cost.

**Questions:**

My questions are in parallel with the weaknesses I listed above:
- Could you please motivate more about the advantages of utilizing predictions in the offline setting with practical applications?
- How does your setup differ from the works of [Bampis et al., 2024] and [Ghoshal et al., 2024]?
- Do you think it is possible to define a computational upper bound on the cost of getting the predictions so that an approximation algorithm can still be considered polynomial? In other words, can there be a trade-off between the cost of predictions and the improvement on the worst-case bound?

**Limitations:**

Limitations of the paper are not explicitly discussed beyond the assumptions made in Section 2.

---

> ### Author Response · Authors · 2024-08-05
>
> We thank the reviewer for the comments and constructive encouragement.
>
> - *Could you please motivate more about the advantages of utilizing predictions in the offline setting with practical applications?*
>
> Predictions in both the offline and online settings help overcome worst-case outcomes by providing additional information about a specific instance. In the offline setting, predictions help bypass computational lower bounds as against information-theoretic ones online. In practice, predictions for offline problems model different scenarios; we list three examples here:
> They can be generated by machine learning models based on solving similar instances previously. This is common for applications such as revenue optimization in auctions.
> They can be used to model advice received from domain experts.
> They can be used to model settings in which a first, coarse solution has already been computed. An algorithm can take advantage of this information to provide a “warm start” without repeating the possibly expensive computation.
>
> - *How does your setup differ from the works of [Bampis et al., 2024] and [Ghoshal et al., 2024]?*
>
> The parallel work of [Bampis et al] considers dense instances of certain maximization problems, including max-cut. Their notion of dense instances have $\Omega(n^2)$ edges (whereas our max-cut results are for general instances). This high density allows them to use advice from only a poly-logarithmic number of calls to the oracle. Moreover, their focus is improving the running time of the PTAS algorithms, whereas our focus is on getting improvements to the approximation guarantees.
>
> The independent and unpublished work of [Ghoshal et al.] considers a model very similar to ours. The original version of their work (concurrent to ours) considered dense cases of max-cut and the closely related max-2LIN. Their notion of density is much weaker than Bampis et al, and is similar to our notion of wide instances. A more recent version (subsequent to ours, and uploaded last week) of their paper gives algorithms for dense instances of 3LIN (and hardness for 4LIN), in contrast to our work that goes in the direction of exploring more general 2CSPs.
>
> - *Do you think it is possible to define a computational upper bound on the cost of getting the predictions so that an approximation algorithm can still be considered polynomial? In other words, can there be a trade-off between the cost of predictions and the improvement on the worst-case bound?*
>
> The computational hardness of Max-Cut (in particular the Unique-Games hardness) means that any polynomial-time approximation algorithm would have an approximation guarantee no better than $\alpha_{GW}$. Hence, we cannot restrict the algorithm to polynomial computation in a standard sense. It would definitely be interesting to consider a model with “parsimonious predictions” where we count the number of queries (say $Q$) made to the prediction oracle, and then study the approximation factor as a function of both the parameter $\varepsilon$ and $Q$. Note that since we are counting calls to an oracle instead of computation time, the hardness results for polynomial-time algorithms are no longer relevant in this setting. The parallel work of Bampis et al. considers parsimonious predictions for dense instances of max-cut with quadratically many edges, and it may also be interesting to consider it for general instances; this seems like a cool direction to explore!

---

### Decision · Program_Chairs · 2024-09-25

**Decision:**

Accept (poster)

**Comment:**

This submission introduces a new model of advice for Max-Cut and similar CSPs.
They design algorithms that can take advantage of such advice.

The reviewers generally like the model and result, but there was a strong sentiment that the authors' title about generic "maximization problems" is a bit misleading. I agree and strongly encourage the authors to change their title to explicitly say Max-Cut / CSPs for their camera ready.